# The Power of Linear Combinations: Learning with Random Convolutions

## Abstract

Following the traditional paradigm of convolutional neural networks (CNNs), modern CNNs manage to keep pace with more recent, for example, transformer-based, models by not only increasing model depth and width but also the kernel size. This results in large amounts of learnable model parameters that need to be handled during training. While following the convolutional paradigm with the according spatial inductive bias, we question the significance of *learned* convolution filters. In fact, our findings demonstrate that many contemporary CNN architectures can achieve high test accuracies without ever updating randomly initialized (spatial) convolution filters. Instead, simple linear combinations (implemented through efficient $1 \times 1$ convolutions) suffice to effectively recombine even random filters into expressive network operators. Furthermore, these combinations of random filters can implicitly regularize the resulting operations, mitigating overfitting and enhancing overall performance and robustness. Conversely, retaining the ability to learn filter updates can impair network performance. Finally, although the improvement we see from learning $3 \times 3$ convolutions is relatively small, the learning gains increase proportionally with kernel size. We attribute this to the independently and identically distributed (*i.i.d.*) nature of default initialization schemes.

## 1 Introduction

Convolutional Neural Networks (CNN) are building the backbone of state-of-the-art neural architectures in a wide range of learning applications on $n$-dimensional array data, such as standard computer vision problems like 2D image classification (Cai et al., 2023; Liu et al., 2022; Brock et al., 2021), semantic segmentation (Wang et al., 2022; Cai et al., 2023), or scene understanding (Berenguel-Baeta et al., 2022). In order to solve these tasks, modern CNN architectures are learning the weights of millions of convolutional filter kernels. This process is not only very compute and data-intensive, but apparently also mostly redundant as CNNs are learning kernels that are bound to the same distribution, even when training different architectures on different datasets for different tasks (Gavrikov & Keuper, 2022a), or can be replaced by random substitutes (Zhou et al., 2019). Yet if - in oversimplified terms - all CNNs are learning the "same" filters, one could raise the fundamental question if we actually need to learn them at all. In this realm, several works have attempted to initialize convolution filters with better resemblance to converged filters, e.g., Yosinski et al. (2014); Trockman et al. (2023).

Contrarily, in order to investigate if and how the training of a CNN with non-learnable filters is possible, we retreat to a simpler setup that eliminates any possible bias in the choice of the filters: we simply set random filters. This is not only practically feasible since random initializations (Glorot & Bengio, 2010; He et al., 2015a) of kernel weights are part of the standard training procedure, but also theoretically justified by a long line of prior work investigating the utilization of random feature extraction (e.g., see (Rahimi & Recht, 2007) for a prominent example) prior to the deep learning era.

One cornerstone of our analysis is the importance of the pointwise ($1 \times 1$) convolution (Lin et al., 2014), which is increasingly used in modern CNNs. Despite its name and similarities in the implementation details, we will argue that this learnable operator differs significantly from spatial $k \times k$ convolutions and learns linear combinations of (non-learnable random) spatial filters.

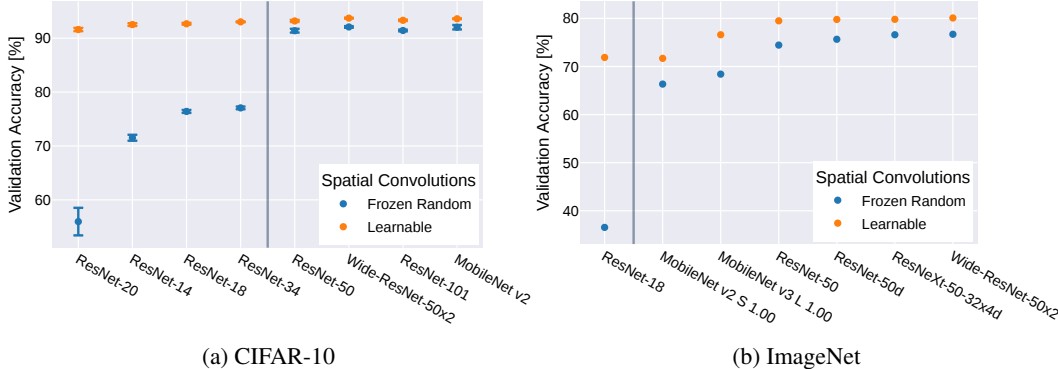

(a) CIFAR-10          (b) ImageNet

Figure 1: Validation accuracy of different **off-the-shelf models** trained on **(a)** CIFAR-10 and **(b)** ImageNet with random frozen random vs. learnable spatial convolutions. Models right of the vertical divider use blocks that integrate $1 \times 1$ convolutions after spatial convolutions and are, therefore, able to construct expressive filters from linear combinations of random filters. CIFAR-10 and ImageNet results are reported over 4 and 1 run(s), respectively.

In this paper, we deliberately avoid introducing new architectures. Instead, we take a step back and analyze an important operation in CNNs: *linear combinations*. We summarize our key contributions as follows:

- We show that modern CNNs (with specific $1 \times 1$ configurations serving as linear combinations) can be trained to high validation accuracies on computer vision tasks without ever updating weights of randomly initialized spatial convolutions, and, we provide a theoretical explanation for this phenomenon.

- By disentangling the linear combinations from intermediate operations, we empirically show that at a sufficiently high rate of those, training with random filters can outperform the accuracy and robustness of fully learnable convolutions due to implicit regularization in the weight space. Alternatively, training with learnable convolutions and a high rate of linear combinations decreases accuracy.

- Based on our observations, we conclude that methods that seek to initialize convolution filters to accelerate training must consider linear combinations during benchmarks. For the common $3 \times 3$ filters, there are only small margins for optimization compared to random baselines. Yet, current random methods struggle with larger convolution kernel sizes, due to their uniform distribution in the spatial space. As such, it is to be expected that better initializations can be found there.

## 2    RELATED WORK

**Random model parameters.** Modern neural network weights are commonly initialized with values drawn *i.i.d.* from uniform or normal distributions with the standard deviation adjusted according to the channel fan, based on proposed heuristics by He et al. (2015a); Glorot & Bengio (2010) to improve the gradient flow (Hochreiter, 1991; Kolen & Kremer, 2001). Rudi & Rosasco (2017) provided an analysis of generalization properties of such random neural networks and conclude that many problems exist, where exploiting random features can reach a significant accuracy, at a significant reduction in computation cost. Indeed, Ulyanov et al. (2018) demonstrated that randomly weighted CNNs can provide good priors for standard inverse problems such as super-resolution, inpainting, or denoising. Additionally, Frankle et al. (2021) showed that only training $\beta$ and $\gamma$ parameters of Batch-Normalization layers (Ioffe & Szegedy, 2015) results in a highly non-trivial performance of CNN image classifiers, although only affine transformations of random features are learned. But even when training all parameters, certain layers seem to learn negligible representations: Zhang et al. (2022) show that entire weights of specific convolution layers can be reset to *i.i.d.* initializations *after training* without significantly hurting the accuracy. The number of affected layers depends on the specific architecture, parameterization, and problem complexity. Finally, both Zhou et al. (2019);

Ramanujan et al. (2019) demonstrated that sufficiently large randomly-initialized CNNs contain subnetworks that achieve good (albeit well below trained) performance on complex problems such as ImageNet. Both approaches apply unstructured weight pruning to find these subnetworks and are based on the *Lottery Ticket Hypothesis* (LTH) (Frankle & Carbin, 2019) which suggests that deep neural networks contain extremely small subgraphs that can be trained to the same accuracy as the entire network.

**Learning convolution filters from base functions.** A different line of work explores learning filters as linear combinations as different (frozen) bases (Hummel, 1979) such as bitmaps (Juefei-Xu et al., 2017; Wan et al., 2018), DCT (Ulicny et al., 2022), Wavelets (Liu et al., 2019), Fourier-Bessel (Qiu et al., 2018), eigenimages of pretrained weights (Tayyab & Mahalanobis, 2019), or low-rank approximations (Jaderberg et al., 2014). Our analysis can be seen as a baseline for these works. However, most bases-approaches enforce the same amount of filters in every layer, whereas, naturally, the amount of filters varies per layer (as defined by the architecture). Furthermore, the number of bases is finite, which limits the amount of possible linear combinations. Contrary, there are infinitely many random filters. This "overcompleteness" may in fact be necessary to train high-performance networks as suggested by the LTH (Frankle & Carbin, 2019).

**Analysis of convolution filters.** Multiple works have studied learned convolution filters, e.g., Yosinski et al. (2014) studied their transferability and (Madry et al., 2018; Gavrikov & Keuper, 2022b) analyzed the impact of adversarial training on those. A long thread of connected research (Olah et al., 2020a;b;c; Cammarata et al., 2020; 2021; Schubert et al., 2021; Voss et al., 2021a;b; Petrov et al., 2021) extensively analyzed the features, connections, and their organization of a trained InceptionV1 (Szegedy et al., 2014a) model. Among others, the authors claim that different CNNs will form similar features and circuits even when trained for different tasks. The findings are replicated in a large-scale analysis of learned $3 \times 3$ convolution kernels (Gavrikov & Keuper, 2022a), which additionally reveals that CNNs generally seem to learn highly similar convolution kernel pattern distributions, independent of training data or task. Further, the authors find that the majority of kernels seem to be randomly distributed or defunct, and only a small rate seems to be performing useful transformations.

**Pointwise convolutions.** Lin et al. (2014) first introduced the concept of *network in network* in which pointwise ($1 \times 1$) convolutions are used to "enhance the model discriminability for local receptive fields". Although implemented similarly to spatial convolutions, pointwise convolutions do not aggregate the local neighborhood but instead compute linear combinations of the inputs and can be seen as a kind of fully-connected layer rather than a traditional convolution. Modern CNNs often use pointwise convolutions (e.g., (He et al., 2015b; Sandler et al., 2018; Liu et al., 2022)) to reduce the number of channels before computationally expensive operations such as spatial convolutions or to approximate the computation of regular convolutions using depthwise filters (*depthwise separable convolutions* (Chollet, 2017)). Alternatively, pointwise convolutions can also be utilized to fully reparameterize spatial convolutions (Wu et al., 2018).

## 3  PRELIMINARIES

We define a 2D convolution layer by a function $\mathcal{F}(X; W)$, $\mathcal{F}$ transforming an input tensor $X$ with $c_{\text{in}}$ input-channels into a tensor with $c_{\text{out}}$ output-channels using convolution filters with a size of $k_0 \times k_1$. Without loss of generality, we assume square kernels with $k = k_0 = k_1$ in this paper. Further, we denote the learned weights by $W \in \mathbb{R}^{c_{\text{out}} \times c_{\text{in}} \times k \times k}$. The outputs $Y_i$ are then defined as $Y_i = W_i * X = \sum_{j=1}^{c_{\text{in}}} W_{i,j} * X_j$ for $i \in \{1, \ldots, c_{\text{out}}\}$. Note how the result of the convolution is reduced to a linear combination of inputs with a now **scalar** $W_{i,j}$ for the special case of $k = 1$ (pointwise convolution): $Y_i = \sum_{j=1}^{c_{\text{in}}} W_{i,j} * X_j = \sum_{j=1}^{c_{\text{in}}} W_{i,j} \cdot X_j$.

We assume *Kaiming Uniform* (He et al., 2015a) as the initialization of model weights (default in PyTorch (Paszke et al., 2019)) but also compare to other methods later. Here, every kernel weight $w \in W$ is drawn *i.i.d.* from a uniform distribution bounded by a heuristic derived from the *input fan* (inputs $c_{\text{in}} \times$ kernel area $k^2$). At default values, this is equivalent to $w \sim \mathcal{U}_{[-a,a]}$ with $a = 1/\sqrt{c_{\text{in}}k^2}$.

## 4 SOME CNNS PERFORM WELL WITHOUT LEARNED CONVOLUTIONS

First, we analyze the performance of different CNNs that vary in depth, width, and implementation of the convolution layer, when the spatial convolution weights are fixed to their random initialization. For simplicity, we will refer to such models as ***frozen random*** through the remainder of the paper. Pointwise convolutions and all other operations always remain learnable.

After training common off-the-shelf architectures such as (CIFAR)-ResNets (He et al., 2015b), Wide-ResNet-50x2 (Zagoruyko & Komodakis, 2016), and MobileNet v2 (Sandler et al., 2018) on CIFAR-10 (Krizhevsky, 2009) we notice an interesting difference in the performance (Figure 1a). Although all models achieve an approximately similar validation accuracy when trained normally, we observe two kinds of frozen random behavior: ResNet-50/101, Wide-ResNet-50x2, and MobileNet v2 show only minor drops in accuracy (1.6-1.9% difference), while the other models show heavy drops of at least 16%. We obtain similar observations on ImageNet (Deng et al., 2009) training ResNet-18/50 (He et al., 2015b), ResNet-50d (He et al., 2019), ResNeXt-50-32x4d (Xie et al., 2017), Wide-ResNet-50x2 (Zagoruyko & Komodakis, 2016), and MobileNet v2 S/v3 L (Howard et al., 2019) (Figure 1b). Accordingly, all models except ResNet-18 converge with "just" 3.21-8.19% difference in validation accuracy. Instead, ResNet-18 shows a gap of 35.34%. We also find that increasing depth [R-18 vs. R-34] or width [R-50 vs. WRN-50x2], as well as reducing the kernel size in the stem [R-50 vs. R-50d] decreases the gap between frozen random and normal training.

An explanation for the performance differences can be found in the architectures: models, where we observe smaller gaps, use Bottleneck-Blocks (He et al., 2015b) (or variants thereof (Sandler et al., 2018)) which process the outputs of spatial convolution layers with pointwise ($1 \times 1$) convolutions. Neglecting intermediate operations, the linear nature of convolutions lets us reformulate the operation:

**Lemma 4.1.** *A pointwise convolution applied to the outputs of a spatial convolution layer is equivalent to a convolution with linear combinations (LCs) of previous filters with the same weights.*

*Proof.* Assume that the $l$-th layer is a spatial convolution with $k > 1$, and inputs into a $k = 1$ pointwise convolution layer $(l + 1)$ with $X^{(l)}$ being the input to the $l$-th layer. Then combining the equations from Section 3 results in:

$$Y_i^{(l+1)} = \sum_{j=1}^{c_{in}^{(l+1)}} W_{i,j}^{(l+1)} \cdot X_j^{(l+1)} = \sum_{j=1}^{c_{in}^{(l+1)}} W_{i,j}^{(l+1)} \cdot \left( W_i^{(l)} * X^{(l)} \right) = X^{(l)} * \sum_{j=1}^{c_{in}^{(l+1)}} \left( W_{i,j}^{(l+1)} \cdot W_i^{(l)} \right)$$

(1)

As such, a set of (sufficiently many) random filters can be transformed into any other (useful) filter by learnable LCs (for an example see Figure 2). Contrary, models without linear combinations (such as Basic-Block ResNets) are restricted to the transformations that random filters produce, and, therefore will show lower accuracies. However, real implementations may include non-linear intermediate operations such as ReLUs and, thus, only approximately behave as in Equation (1).

Figure 2: Example of LCs of random filters resulting in more expressive filters.

Based on our observations, we hypothesize that (sufficiently large) CNNs exist, that despite being trained with random frozen filters will achieve a comparable accuracy to training of all parameters. In the following sections, we will prove this and explore further properties as well as limitations of these networks.

## 5 EXPLORATION OF LINEAR COMBINATIONS

To systematically study the properties of linear combinations of random filters free of the interference of potentially non-linear intermediate operations, we experiment with adjusted networks. Therein, we replace every spatial convolution layer with an *LC-Block*. This block is designed to allow us to control the linear combination rate of convolution filters without affecting other layers, but also introduces linear combinations to networks that didn't include them previously. We implement the block by replacing a spatial convolution with $c_{in}$ input channels and $c_{out}$ output channels with a combination of

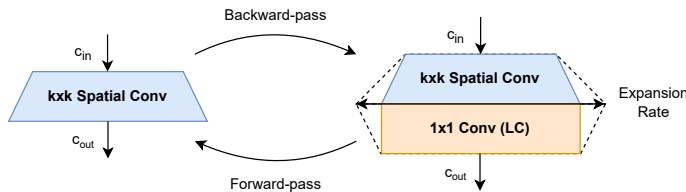

Figure 3: **LC-Block**: Appending a pointwise convolution layer to all spatial convolution layers in the networks allows controlling the linear combination rate without altering the number of outputs via an expansion factor $E$. The LC-Block is a reparameterization of the original layer.

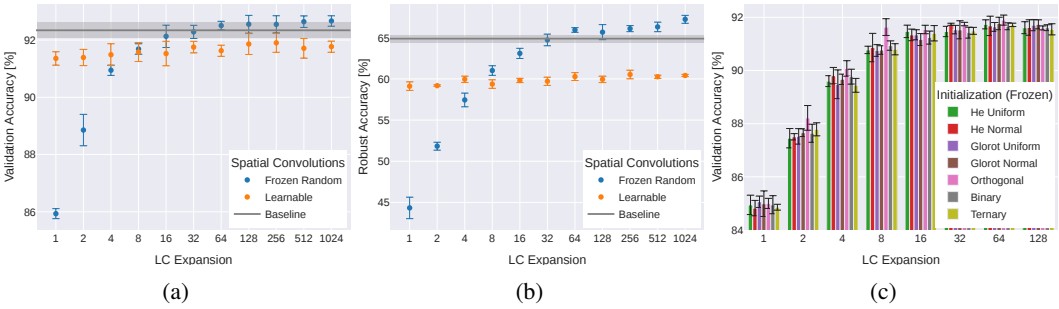

Figure 4: Validation accuracy of ResNet models trained on CIFAR-10 under **increasing LC expansion** ($D = 20, W = 16$). **(a)** clean accuracy vs. expansion, **(b)** robust accuracy against light adversarial attacks vs. expansion, **(c)** different initializations. After sufficiently many linear combinations, **frozen random models outperform the baseline and learnable LC models**. Results are reported over 4 runs.

a spatial convolution with $c_{\text{out}} \times E$ filters which are directly fed into a pointwise convolution with $c_{\text{out}}$ outputs. Intermediate operations such as activations or normalization layers between the two layers are deliberately omitted. Following Equation (1), the LC-Block is a *reparameterization* of the original convolution layer that can be folded back into a single spatial convolution operation during the forward-pass (Figure 3). We will refer to the reparameterization into a single weight as *combined filters* in this paper. Because of this, neither introducing the LC-Block nor changing the expansion rate increases the expressiveness of the network but affects the learning process. Further, increasing the expansion rate only adds a marginal overhead to the training time.

Exemplarily, we study this modification to the (Basic-Block) CIFAR-ResNet as introduced in (He et al., 2015b). Compared to regular ResNets, CIFAR-ResNets have a drastically lower number of parameters and are, therefore, more suitable for large-scale studies such as the one presented in the following sections. We denote modified models by *ResNet-LC-{D}-{W}x{E}*, where $D$ is the network depth, i.e., the number of spatial convolution and fully-connected layers, $W$ the network width (default 16), i.e., the initial number of channels communicated between Basic-Blocks, and $E$ the LC expansion factor (default 1). In principle, LC-Blocks can be applied to any CNN architecture, yet they are unlikely to be relevant outside this study and just serve as a proxy for the LCs at various expansions that naturally happen in off-the-shelf architectures and allows us to study different expansion rates without having to find specific architectures.

## 5.1 INCREASING THE RATE OF LINEAR COMBINATIONS

As per our hypothesis in Section 4, an increase in LCs should eventually close the gap between frozen random and learnable spatial convolutions. We test this by exponentially increasing the expansion factor of ResNet-LC-20-16 trained on CIFAR-10 and benchmark the performance of frozen random and learnable ResNet-LC against a fully learnable and unmodified ResNet-20-16 baseline.

We find three important observations based on the results in Figure 4: ① The accuracy of **frozen random** models constantly increases with expansion. Surprisingly, at $E = 8$ they **outperform**

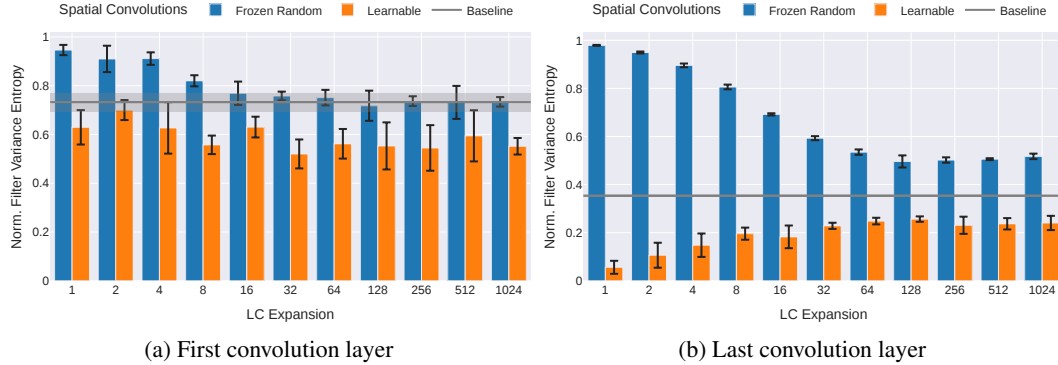

(a) First convolution layer                    (b) Last convolution layer

Figure 5: **Filter variance entropy** normalized by the randomness threshold as a metric of diversity in filter patterns of the **(a)** first and **(b)** last layer. Measured on ResNet-LC-20-16x$\{E\}$ trained on CIFAR-10 with frozen random or learnable spatial convolutions under increasing **LC expansion** $E$. Values $\geq 1$ indicate a random distribution of kernel patterns, while values of 0 indicate a collapse to one specific pattern. Results are reported over 4 runs.

**learnable ResNet-LCs**, and, at $E = 32$, they even **outperform the baseline**. Beyond that, the accuracy appears to stagnate; ② Similarly, we observe an **increase in robustness** to light adversarial attacks (Szegedy et al., 2014b) ($\ell_\infty$-FGSM (Goodfellow et al., 2015) with $\epsilon = 1/255$) of frozen random models with expansion. Eventually, they again outperform the learnable and baseline counterparts. However, unlike clean validation accuracy, the robust accuracy does not appear to saturate under the tested expansion rates; ③ **Learnable ResNet-LCs perform worse than the original baseline**. The gap diminishes with expansion but even at the highest tested expansion, the performance remains at the lower end of the baseline;

**What differences can be observed in representations?** Due to the clean and robust accuracy gap, it seems viable to conclude that frozen random models learn different representations. Compared to the baseline, random frozen models are intrinsically limited in the patterns that the combined filters can learn but this space increases with the number of linear combinations and matches the baseline at infinite combinations. At the same time, this limitation also serves as regularization and prevents overfitting which may explain why random frozen models generalize better than the baselines.

To further investigate this we measure the *filter variance entropy* (Gavrikov & Keuper, 2022a) of the initial and final convolution layers (Figure 5). This singular value decomposition-based metric measures the diversity of filter kernel patterns, by providing a measurement in an interval between entirely random patterns (as seen in just initialized weights; a value $\geq 1$) and a singular pattern repeated throughout all kernels (a value of 0).

In the first layer of the baseline model, we observe an expected balanced diversity of patterns (compare to (Yosinski et al., 2014)) that is not random but neither highly repetitive. Whereas, the learnable LC models show a significantly lower diversity there that remains relatively stable independent of the expansion rate. Random frozen LC models are initially very close to random distributions but the diversity decreases with expansion and eventually converges slightly below the baseline but well above the diversity of learnable LC models. We show visualizations of the initial layer in Appendix A.5.

In the last layer, we observe a significantly lower baseline due to degenerated convolution filters that collapse into a few patterns that are repeated throughout the layer, as shown in (Gavrikov & Keuper, 2022a). Learnable LC models show an even lower diversity and thus a higher degree of degeneration. In this layer, however, the diversity increases with expansion but still remains well below the baseline. We attribute the improvements with respect to the expansion to the smoothing/averaging effect of large numbers of linear combinations (analogous to the findings about ensemble averaging in (Allen-Zhu & Li, 2023)) which prevent collapsing into one specific pattern but still cannot avoid collapsing in general. For random frozen LC models, we again observe initially highly random filter patterns that decrease fast in diversity with increasing expansion but remain well above the baseline and learnable LC models.

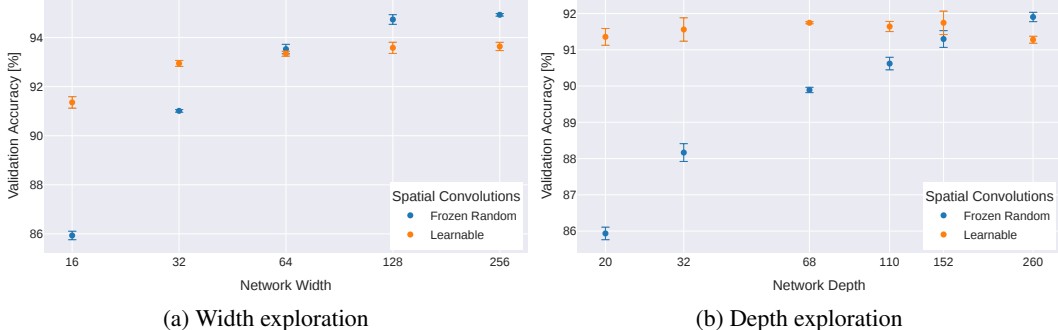

(a) Width exploration                    (b) Depth exploration

Figure 6: Validation accuracy of ResNet-LC models trained on CIFAR-10 with frozen random or learnable spatial convolutions under increasing **(a) network width** ($D = 20, E = 1$), and **(b) network depth** ($W = 16, E = 1$). After sufficiently many linear combinations, **frozen random models again outperform learnable** ones. Results are reported over 4 runs.

Gavrikov & Keuper (2022a) have linked poor filter diversity to overfitting. Based on the diversity metrics it is thus not surprising that the learnable LC models underperform the baseline. In contrast to that, random frozen LC models cannot collapse that easily and remain even more diverse than the baseline. Further, Gavrikov & Keuper (2022b) have attributed (adversarial) robustness with a higher diversity of filter patterns which correlates with our findings of higher robustness in random frozen LC models.

It is worth noting, that the shown improvements in robustness of random frozen networks are more of academic nature and are no match against special techniques such as adversarial training (Madry et al., 2018) and cannot withstand high-budget attacks.

**Other options to increase the LC rate.** An explicit LC expansion as in our LC-Block is not the only way to increase the rate of linear combinations in a network already containing LCs. Generally, increasing the network width naturally increases the number of LCs and closes the accuracy gap (Figure 6a; break-even at approx. $W = 64$). In addition, the gap diminishes with increasing depth due to the compositional nature of deep neural networks (Figure 6b; break-even at approx. $D = 260$). Note that we use a ResNet-LC-20-16x1 here as a proxy for any networks containing linear combinations, and thus, the learnable LC model is the baseline. Similar trends could be observed with unmodified Bottleneck-Block ResNets.

**Other initializations.** Alternating different common *i.i.d.* initializations, such as Glorot & Bengio (2010) and He et al. (2015a) from uniform and normal distributions, and random binary (Juefei-Xu et al., 2017) or ternary (Wan et al., 2018) filters we do not observe notable differences in performance (Figure 4c). As the filters are *i.i.d.*, there is no guarantee that they would span a basis that would allow combining the random filters into useful filters and thus match the performance of learnable models. As the expansion increases, the odds of forming such a basis also increase, independent of the actual methods. Once a sufficient expansion has been reached, the basis becomes overcomplete and the gains stagnate. Yet, switching to (semi-)orthogonal inits (Saxe et al., 2014) we notice gains for small expansion rates. Obviously, orthogonal initializations accelerate this process by guaranteeing a basis after sufficiently many filters, despite in general not being a preferred choice for fully learnable models. We expect more complex inits to even further improve performance.

## 5.2 Scaling to other Problems

In this section, we aim to demonstrate that our findings also transfer to problems, such as CIFAR-100 (Krizhevsky, 2009), SVHN (Netzer et al., 2011), Fashion-MNIST (Xiao et al., 2017). The results in Table 1 confirm our previous findings. At $E = 128$ random frozen ResNet-LC-20-16 models perform better or on par with the baseline, while the learnable LC variants perform worse. Additionally, we also perform experiments on ImageNet (Deng et al., 2009). Since the ResNet-20 architecture (He et al., 2015b) is under-parameterized for this problem we switch to the more powerful ResNet-18d architecture (He et al., 2019) which also avoids kernels larger than $3 \times 3$. Again, we observe a similar behavior (Table 2). We were not able to outperform the baseline, but we attribute this to the

granularity of tested expansion rates and the fluctuations in the measurements of a single run. In Appendix A.1, we also demonstrate that our findings transfer to Object Detection and Segmentation problems.

Table 1: Validation accuracy of a ResNet-20-16 trained on **multiple other datasets** as baseline and as learnable or random frozen LC with an expansion rate of 128. Results are reported over 4 runs. **Best**, second best.

| Dataset | Validation Accuracy [%] | | |
| --- | --- | --- | --- |
| | Baseline | LC-Learn. | LC-Frzn.Rand. |
| CIFAR-10 | 92.34±0.24 | 91.86±0.34 | **92.55±0.28** |
| CIFAR-100 | 63.91±0.37 | 62.51±0.45 | **64.72±0.47** |
| Fashion-MNIST | 93.50±0.15 | 93.25±0.14 | **93.65±0.11** |
| SVHN | **96.38±0.12** | 96.16±0.04 | 96.27±0.14 |

Table 2: Validation accuracy of a ResNet-18d trained on **ImageNet**. Learnable LCx128 exceeded 4x A100 VRAM. Results are reported over 1 run.

| Model | Frozen Random | | Learnable | |
| --- | --- | --- | --- | --- |
| | Top-1 | Top-5 | Top-1 | Top-5 |
| Baseline | 37.85 | 61.37 | **72.60** | **90.69** |
| LCx1 | 61.28 | 83.11 | 71.82 | 90.44 |
| LCx8 | 71.30 | 90.03 | 71.88 | 90.40 |
| LCx64 | 72.46 | 90.57 | 71.94 | 90.41 |
| LCx128 | 72.31 | 90.64 | *n/a* | |

## 5.3 THE ROLE OF THE KERNEL SIZE

Our networks use the default $3 \times 3$ kernel size, which was dominant for the past years. However, recently proposed CNNs often increase the kernel size (Tan & Le, 2020; Liu et al., 2022; Trockman & Kolter, 2023), sometimes to as large as $51 \times 51$ (Liu et al., 2023). To verify that our observations hold on larger kernels, we increase the convolution sizes in a ResNet-LC-20-16 to $k \in \{5, 7, 9\}$ (with a respective increase of input padding) and measure the performance gap between random frozen and learnable spatial convolutions on CIFAR-10 (Krizhevsky, 2009). Our results (Figure 7a) show that the gap between frozen random and regular models significantly increases with kernel size, but again steadily diminishes with increasing expansion and eventually breaks even for all our tested expansions.

Obviously, from a combinatorial perspective, more linear combinations are necessary to learn a specific kernel as the kernel size becomes larger. To understand further differences we take a closer look at the combined filters and compare random frozen against learnable models. We find an important difference there: (large) learnable filters primarily learn the weights in the center of the filter. Outer regions remain largely constant. This effect is barely or not at all visible for $3 \times 3$ kernels but gradually manifests with increasing kernel size. To better capture this effect we visualize these findings in the form of heatmaps highlighting the variance for each filter weight (Figure 7b). We compute the variance over all convolution kernels (total number $N$) in a model. First, the kernels are normalized by the standard deviation of the entire convolution weight in their respective layer and finally stacked into a 3D tensor of shape $k \times k \times N$ on which we compute the variance over the last axis. In the resulting heatmaps, we notice that random frozen models do not match this spatial distribution - their variance heatmaps are uniformly distributed independently of the kernel size. As such, the differences between learnable and random frozen models increase with kernel size due to poor reconstruction ability which correlates with the increasing accuracy gap. The cause for the uniform variance distribution can be found in the initialization. Initial kernel weights are drawn from *i.i.d.* initializations (Glorot & Bengio, 2010; He et al., 2015a) without consideration of the weight location in the filter. Linear combinations of these filters thus remain uniformly distributed and cannot manage to learn sharp patterns. For visualizations of learnable and combined frozen random filters refer Appendix A.5.

## 6 CONCLUSION, DISCUSSION, AND OUTLOOK

In our controlled experiments we have shown that **random frozen CNNs can outperform fully-trainable baselines** whenever a sufficient amount of linear combinations is present. The same findings apply to many modern real-world architectures which implicitly compute linear combinations (albeit not as clean due to intermediate operations).

In such networks, **learned spatial convolution filters only marginally improve the performance compared to random baseline**, and, in the settings of very wide, deep, or networks with an otherwise large number of linear combinations, learning spatial filters does not only result in no improvements

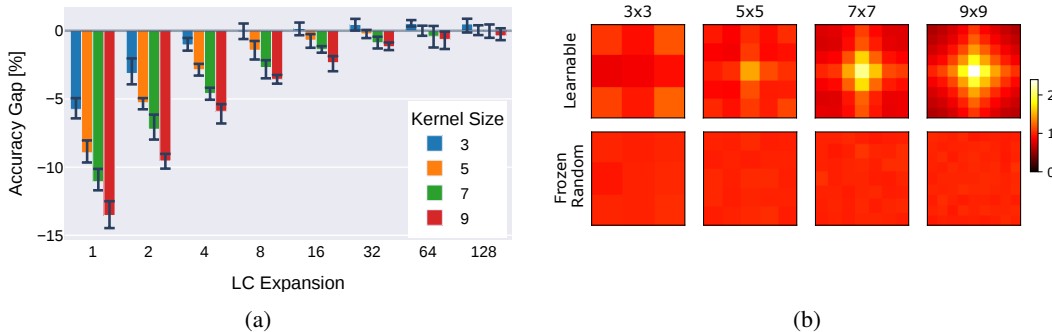

(a)                                                                              (b)

Figure 7: **Experiments with larger kernel sizes. (a) Gap in validation accuracy** between frozen random and learnable ResNet-LC-20-16x$\{E\}$ on CIFAR-10 with different convolution kernel sizes under increasing LC expansion $E$. Results are reported over 4 runs. **(b) Spatial variance in the weights of combined filters** of learned (top row) and frozen random (bottom row) models.

but may even hurt the performance and robustness while wasting compute resources on training these parameters. This is in line with the findings (Zhang et al., 2022) that CNNs contain (spatial) convolution layer that can be reset to their *i.i.d.* initialization without affecting the accuracy and the number of such layers increases in deeper networks. In contrast, **training with random frozen filters regularizes the effective filter patterns and prevents overfitting**.

Additionally, this implies that works seeking better convolution filter initializations are limited by narrow optimization opportunities in modern architectures and must consider the presence of linear combinations when reporting performance. **Only, under increasing kernel size, learned convolution filters begin to increase in importance** - however, most likely not due to highly specific patterns, but rather **due to a different spatial distribution of weights that cannot be reflected by *i.i.d.* initializations**. It remains to be shown in future work, whether simply integrating this observation into current initialization techniques is sufficient to bridge the gap, though being likely as some proposed alternative initializations, e.g., Trockman et al. (2023) only show improvements for larger kernel sizes.

We also see some cause for concern based on our results. There is a trend to replace spatial convolution operations via pointwise ones whenever possible due to cheaper cost (Sandler et al., 2018; Howard et al., 2019; Liu et al., 2022). Inversely, only a few works focus on spatial convolutions, e.g., promising directions are the strengthening kernel skeletons (Ding et al., 2019; 2022; Vasu et al., 2023), or (very) large kernel sizes (Ding et al., 2022; Liu et al., 2023). However, we have seen that **adding learnable pointwise convolutions immediately after spatial convolutions decreased performance and robustness**. This raises the question of whether this applies to (and limits) existing architectures. One indication for the imperfection of excessive LC architectures may be that VGG-style (Simonyan & Zisserman, 2015) networks (not containing LCs) can be trained to outperform ResNets (Ding et al., 2021). Ultimately, answering this question will require a better understanding of the difference in learning behavior. While we have seen the outcome (filters are becoming less diverse which correlates with overfitting (Gavrikov & Keuper, 2022a) and decreased robustness (Gavrikov & Keuper, 2022b)), it remains unclear as to what the actual cause is. Apparently, it must be linked to the backward-pass, as the forward-pass is identical, e.g., consider a ResNet and an identical twin that inserts LC-Blocks and retains the spatial convolution weights. Setting the pointwise weights in the LC-Block with identity matrices will result in the same representations. Yet, although the network could learn this, it learns an arguably worse representation.

**Limitations of this study.** All our models have been trained using the commonly used (stochastic) gradient descent-based optimization. It is unclear if our findings - in particular the degeneration due to learnable LCs - will transfer to other solvers, such as evolutionary algorithms. Lastly, we have only examined linear combinations through pointwise convolutions. Yet, the same effect is obtainable by spatial kernels that only train the center element as observed in adversarially-trained models (Gavrikov & Keuper, 2022b), fully-connected layers (Rosenblatt, 1958), attention layers (Vaswani et al., 2017), and potentially other operations. We aim to close this knowledge gap in future studies.

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
