# A APPENDIX

## CONTENTS

## A.1 LINEAR COMBINATIONS IN OBJECT DETECTION AND SEGMENTATION TASKS

Additionally to our image classification tasks in the main paper, we now show that our observations of superior frozen random LC performance transfer to other tasks. To this end, we train a YOLO5n detector (Charles, 2013) on the PASCAL VOC 2007 dataset (Everingham et al., 2007) as an example of Object Detection, and U-Nets (Ronneberger et al., 2015) on the *Kaggle Carvana Image Masking Challenge* (Shaler et al., 2017) and *Kaggle LGG Brain MRI Segmentation Challenge* (Mazurowski et al., 2017) datasets as examples of Semantic Segmentation problems. The results in Table 3, Table 4, Table 5 demonstrate that - similar to image classification - frozen random LCs on other problems also eventually outperform fully learnable baselines. On the *Carvana* dataset this already happens at an expansion of 1.

Table 3: Object Detection results with YOLO5n on VOC2007 @ 640 px. Results from a single run.

| Model | mAP@50 | mAP@50:95 | Precision | Recall |
|---|---|---|---|---|
| Learnable Baseline | 70.00 | 43.19 | 71.77 | 65.92 |
| Frozen Baseline | 53.57 | 27.87 | 59.02 | 52.54 |
| Frozen LCx8 | 69.38 | 42.17 | 69.56 | 65.44 |
| Frozen LCx16 | 69.80 | 42.65 | 71.96 | 65.97 |
| Frozen LCx32 | 70.02 | 43.10 | 71.31 | 65.44 |
| Frozen LCx64 | **70.66** | **43.58** | 70.69 | 66.97 |

Table 4: Semantic Segmentation results with U-Net on the Kaggle Carvana Image Masking Challenge (50% resolution). Results from a single run.

| Model | Dice |
|---|---|
| Learnable Baseline | 0.97558 |
| Frozen Baseline | 0.96214 |
| Frozen LCx1 | **0.98814** |

Table 5: Semantic Segmentation results with U-Net on the Kaggle LGG Brain MRI Segmentation Challenge @ 256 px. Results from a single run.

| Model | Dice |
|---|---|
| Learnable Baseline | 0.906556 |
| Frozen Baseline | 0.755624 |
| Frozen LCx2 | 0.896155 |
| Frozen LCx4 | 0.905418 |
| Frozen LCx8 | **0.910441** |

## A.2 TRAINING DETAILS

**Low-resolution datasets.** We train all models for 200 epochs (except in Figures 1a, 4c and 7a and Table 6 where we train for 75 epochs). We use an SGD optimizer (with Nesterov momentum of 0.9) with an initial learning rate of 1e-2 following a cosine annealing schedule (Loshchilov & Hutter, 2017), a weight decay of 1e-2, a batch size of 256, and Categorical Cross Entropy with a label smoothing (Goodfellow et al., 2016) of 1e-1. We pick the last checkpoint for evaluation.
We use the following augmentations:

- **CIFAR-10/100**: For training, images are zero-padded by 4 px along each dimension, apply random horizontal flips, and proceed with $32^2$ px random crops. Test images are not modified.
- **SVHN**: No augmentations.
- **Fashion-MNIST**: Both, train and test images are upscaled to $32^2$ px.

In all cases, the data is normalized by the channel mean and standard deviation.

**ImageNet.** We train all models following Wightman et al. (2021) (A2) with automatic mixed precision training (Micikevicius et al., 2018) for 300 epochs at $224^2$ px resolution without any pre-training and report top-1 and top-5 accuracy for both, learnable and frozen random training. We pick the checkpoint with the highest top-1 validation accuracy for evaluation.

## A.3 DERIVATION OF THE LAYER SCALE COEFFICIENT

We use the default PyTorch initialization of convolution layers: Weights are drawn from a uniform distribution and scaled according to He et al. (2015a). PyTorch uses a default gain of $\sqrt{\frac{2}{1+\alpha^2}}$ with $\alpha = \sqrt{5}$. Which results in $gain = \sqrt{1/3}$. Further, the channel input fan is used for normalization.

The standard deviation for weights drawn from normal distributions is given by:

$$\sigma_{\text{he}} = \frac{\text{gain}}{\sqrt{\text{fan}}} = \frac{\text{gain}}{\sqrt{c_{\text{in}}k^2}} \tag{2}$$

And the standard deviation of a symmetric uniform distribution $\mathcal{U}_{[-a,a]}$ is given by:

$$\sigma = a/\sqrt{3} \tag{3}$$

To retain the standard deviation we, therefore, compute the scaling coefficient as follow:

$$s = \sqrt{3}\sigma_{\text{he}} = \sqrt{3}\frac{\text{gain}}{\sqrt{c_{\text{in}}k^2}} = \sqrt{3}\frac{\sqrt{1/3}}{\sqrt{c_{\text{in}}k^2}} = \frac{1}{\sqrt{c_{\text{in}}k^2}} \tag{4}$$

## A.4 ABLATION OF INTERMEDIATE OPERATIONS IN LC-BLOCKS

Practical CNN architectures often include intermediate operations that influence linear combinations. Exemplarily, we study this in an experiment on ResNet-LC-20-16x64 trained on CIFAR-10 and insert

ReLU, an (affine) BatchNorm operation, and a combination of both between the two convolution layers in an LC-Block. We compare frozen random against learnable models in Table 6. Just adding a BatchNorm layer lowers the performance in both cases. This is somewhat in line with our observations that adding learnable LCs lowered the accuracy as the performed affine transformation is an overparameterization that does not increase expressiveness. Adding a ReLU activation, however, increases the performance due to the additional non-linearity that can be exploited in the combination of filters. In this example, learnable LCs benefit from this and outperform random frozen models, although the random frozen baseline was superior. The combination of both operations performs best in, both, random frozen and learnable models.

Table 6: Influence of intermediate operations in LC-Blocks.

| | **Validation Accuracy** [%] | |
| **Intermediate Operation** | Frozen Random | Learnable |
| --- | --- | --- |
| None | $91.71 \pm 0.08$ | $91.18 \pm 0.10$ |
| ReLU | $92.78 \pm 0.31$ | $93.38 \pm 0.15$ |
| BatchNorm | $91.50 \pm 0.15$ | $91.06 \pm 0.23$ |
| BatchNorm + ReLU | $93.26 \pm 0.18$ | $94.72 \pm 0.16$ |

### A.5 COMBINED FILTERS

Figure 8 shows the combined filters (i. e. the convolutions filters obtained by the linear combination in LC-Blocks) in the first convolutions layers of frozen random and learnable filters at different rates of expansion and for different kernel sizes. The filters of learnable ResNet-LCs remain fairly similar independent of expansion, while the frozen random filters become less random with increasing depth. A well-traceable filter is a green color blob, that evolves from noise to a square blob and eventually to the Gaussian-like filter. Also visible is that larger filters concentrate more of their weights in the center of the filters.

Figure 9 shows the filter variance entropy (FVE) for the same combined filters. Note that contrary to Section 5 we do not normalize the FVE by the randomness threshold, as it was only derived for $3 \times 3$ convolutions by the original authors. Using the non-normalized values, however, allows a comparison independent of kernel size. Once again, we can see that the FVE of learnable LC models remains constant throughout different expansion rates and only marginally decreases with increasing kernel size. For all kernel sizes, we see that frozen random models decrease in FVE at increased expansion. Yet, they are increasingly more diverse with kernel size. Hence, the gap between learnable and frozen random weights significantly increases with increasing kernel size.

We repeat this measurement for the last convolution layer in Figure 10 which shows even larger differences between frozen random and learnable LC models with increasing kernel size. Although we generally observe similar trends, there is one salient difference to the first layer: the diversity of *learnable* LC models collapses for non-$3 \times 3$ layers. This highlights the importance of kernel strengthening (Ding et al., 2019; 2022; Vasu et al., 2023) for larger kernels.

### A.6 POTENTIAL NEGATIVE SOCIETAL IMPACTS

We do not believe that our analysis causes any negative societal impacts. As with most publications in this field, our experiments consumed significant amounts of energy and caused the emission of $CO_2$. However, by exposing non-idealities of current approaches we hope to inspire future researchers to reconsider their network designs to reduce emissions during training.

### A.7 COMPUTATIONAL RESOURCES

The training was executed on internal clusters with *NVIDIA* A100-SXM4-40GB GPUs for a cumulative total of approximately 2901.6 GPU hours. Detailed budgets spent on evaluating the baseline models in Section 4, the linear combination experiments in Section 5 including adversarial evaluation in Section 5.1, ablation of intermediate operations, and abandoned experiments can be found in Table 7.

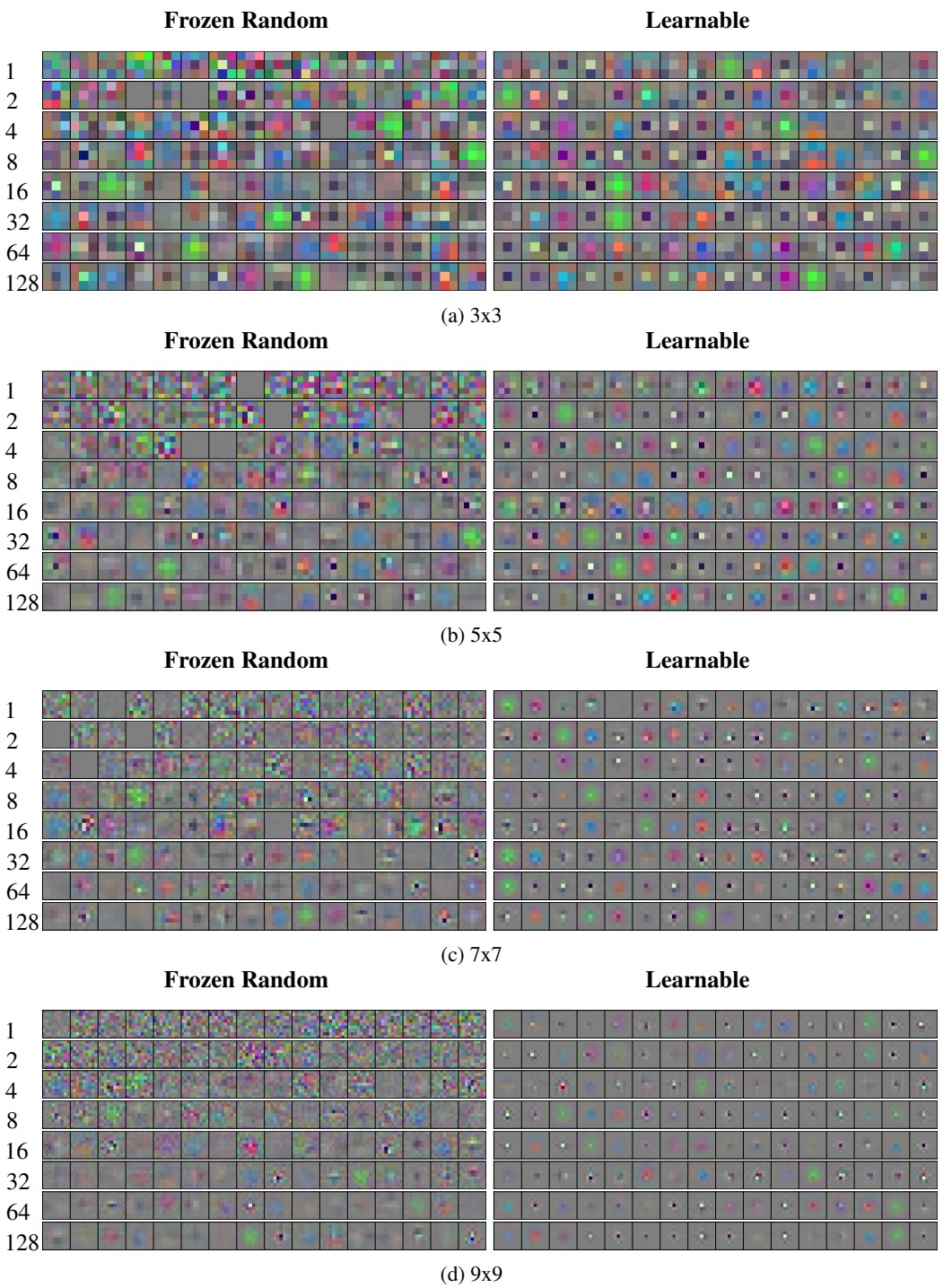

Figure 8: Visualization of the combined filters of the first convolution layer in ResNet-LCs-20-16x$\{E\}$ with increasing expansion $E$ of frozen random (left) or learnable (right) models under different kernel size $k$.

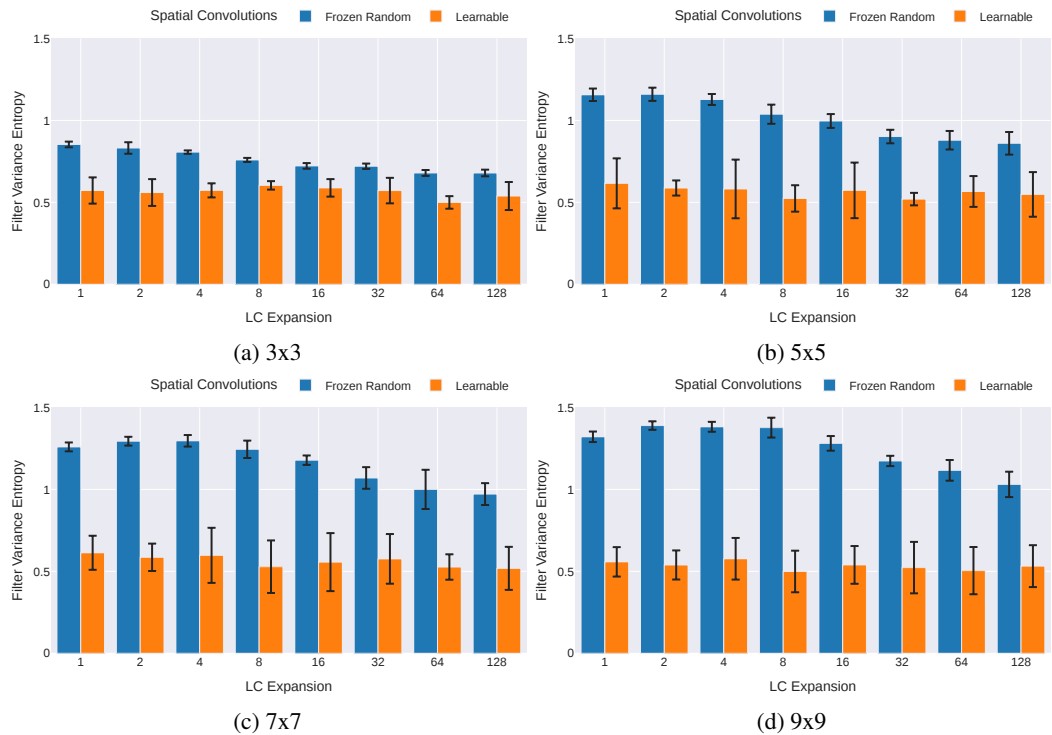

Figure 9: **Variance entropy** (not normalized for comparability) as a measure of diversity in filter patterns of the combined filters in the **first convolution layer** in ResNet-LC-20-16x$\{E\}$ on CIFAR-10. We compare random frozen to learnable models under **increasing LC expansion** $E$ and **kernel sizes** $k$.

Table 7: Detailed compute resources spent for experiments in this paper. Cumulative hours refer to the number of GPUs (*NVIDIA* A100-SXM4-40GB GPUs) used per experiment times the runtime.

| Experiment | Cumulative GPU hours |
|---|---:|
| Baseline ImageNet | 870 |
| Baseline CIFAR-10 | 183 |
| LC CIFAR-10 | 787.3 |
| LC CIFAR-100 | 2.8 |
| LC SVHN | 3.9 |
| LC FashionMNIST | 3.0 |
| LC ImageNet | 869.2 |
| Adversarial Evaluation | 1.3 |
| Ablation | 20.2 |
| Abandoned Experiments | 160.9 |
| **TOTAL** | 2901.6 |

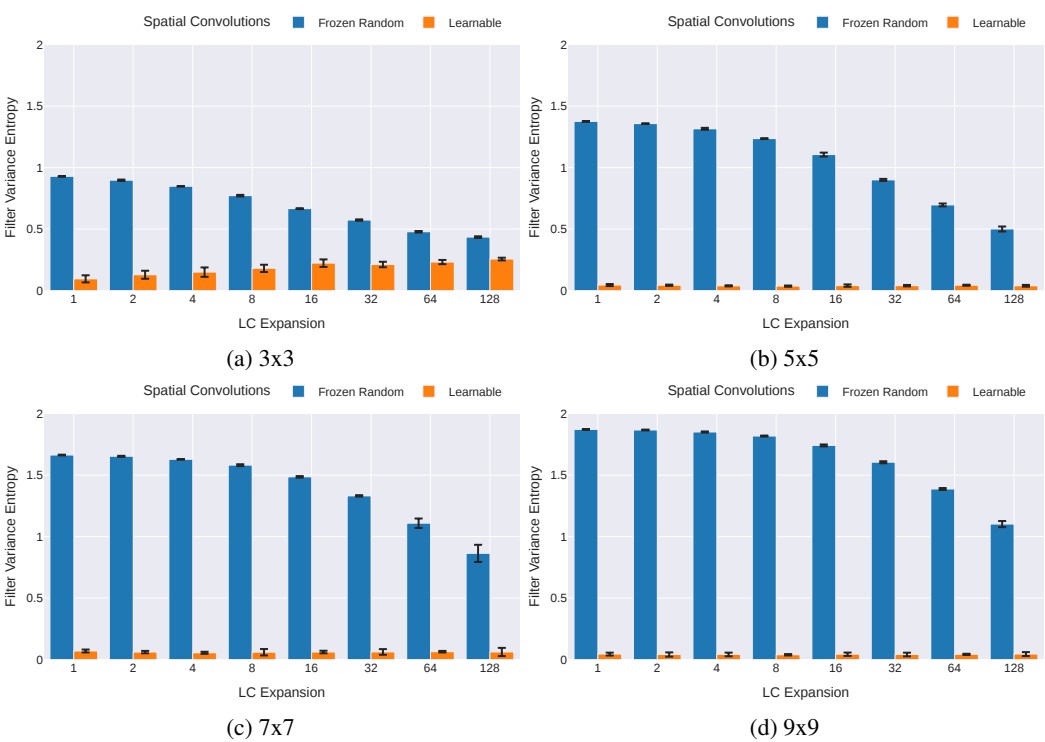

Figure 10: **Variance entropy** (not normalized for comparability) as a measure of diversity in filter patterns of the combined filters in the **last convolution layer** in ResNet-LC-20-16x$\{E\}$ on CIFAR-10. We compare random frozen to learnable models under **increasing LC expansion** $E$ and **kernel sizes** $k$.