# OpenReview forum: "The Power of Linear Combinations: Learning with Random Convolutions"
_ICLR.cc/2024/Conference — Submitted to ICLR 2024_

### Official Review · Reviewer_gGqG · 2023-10-29

**Soundness:** 3 good
**Presentation:** 4 excellent
**Contribution:** 2 fair
**Rating:** 3
**Confidence:** 4

**Summary:**

This paper questions the significance of learned convolution filters, while following the convolutional paradigm with according spatial inductive bias. Contemporary CNN architectures can achieve high test accuracies without updating the randomly initialized filters. It further shows that romdom filters mitigates overfitting and enhances overall performance and robustness. Learning gains increase proportionally with kernel size.

**Strengths:**

- CNNs can be trained to high validation accuracies on computer vision tasks without ever updating weights of randomly intialized spatial convolutions.
- training with random filters can outperform the accuracy and robustness of fully learnable convolutions due to implicit regularization in the weight space.
- The paper is well written.

**Weaknesses:**

- The idea is trivial and well-known.
- the experiments are done on toy models and datasets, which are not convincing.

**Questions:**

- Experiments on state-of-the-art models

---

> ### Author Response · Authors · 2023-11-13
> **Rebuttal**
>
> > W1: The idea is trivial and well-known.
>
> With all due respect, we remind you of the reviewer's instructions for raising weaknesses: *“Be specific, avoid generic remarks. [...] if you believe the contribution lacks novelty, provide references and an explanation as evidence;”*
>
>
> If you believe that our results are well-known, please provide references and explanations as evidence. As a reminder here are our key statements from the conclusion section:
>
> 1. Random frozen CNNs can outperform fully trainable baselines in clean and robust accuracy.
> 2. In modern off-the-shelf networks, learned spatial convolution filters only marginally improve the performance compared to frozen random models in small kernel regimes. The reason for this is an implicit computation of linear combinations between pointwise and spatial filters.
> 3. Adding learnable pointwise convolutions immediately after spatial convolutions can decrease performance and robustness.
> 4. Only, under increasing kernel size, learned convolution filters begin to increase in importance but only due to a different spatial distribution of weights that cannot be reflected by currently practiced i.i.d. initializations.
>
> Needless to say, we disagree that these statements are well-known or trivial.
>
> > W2: the experiments are done on toy models and datasets, which are not convincing.
> > Q1: Experiments on state-of-the-art models
>
> It is not true that we only experiment with toy models and datasets. We ablate the effects on various ResNets scaling linear combinations, width, and depth on CIFAR-10. Following that, we show that the results transfer to multiple other datasets including ImageNet (Sec. 5.2) which is arguably the benchmark in vision, as well as other tasks such as Object Detection and Segmentation in Appendix A.1 containing real-life problems from Kaggle Challenges. Fig. 1 also shows the behavior of popular benchmark CNNs: ResNe(X)t-50 and Wide-ResNet-50. In all of these experiments, we see the same observations - how can the results be unconvincing?
>
> Please let us know if we have addressed your concerns. If you have any other questions/concerns please feel free to ask. We would be grateful if you would reconsider your score in light of our updates. Thank you again for your time and consideration.

---

> > ### Comment · Reviewer_gGqG · 2023-11-14
> >
> > The idea is trivial because:
> > 1. It is well known that separable conv (depthwise + pointwise) is better than naive 3x3 (ref: MobileNet).
> > 2. The implicit computation of linear combination is quite obvious, by basic linear algebra.
> >
> > The experiments are toy.
> > To give you a non-trivial example, on ImageNet, you need to do an experiment with 90%+ baseline accuracy.

---

> > > ### Author Response · Authors · 2023-11-14
> > >
> > > We thank the reviewer for their swift response.
> > >
> > > > It is well known that separable conv (depthwise + pointwise) is better than naive 3x3 (ref: MobileNet).
> > >
> > > There seems to be a misunderstanding - we are not showing that separable convolutions are better. We show that in modern networks you can shift the learning purely to the pointwise convolution *without* ever training spatial convolutions - and without hurting and sometimes even improving the accuracy.
> > >
> > > In fact, none of our main experiments even use depthwise convolutions. However, our findings also apply to depthwise convolutions and, e.g., explain why MobileNet can be trained to similar accuracy (Fig. 1).
> > >
> > > > The implicit computation of linear combination is quite obvious, by basic linear algebra.
> > >
> > > Yes, linear combinations from random bases may be obvious. However, we show that spatial convolutions followed by pointwise convolutions *approximate* a linear combination of their weights and what this means for training (see statements in the initial Rebuttal).
> > >
> > > > The experiments are toy. To give you a non-trivial example, on ImageNet, you need to do an experiment with 90%+ baseline accuracy.
> > >
> > > Please see Tab. 2, where we reached 99.8% of the original ImageNet baseline. On other datasets we even outperform baselines.

---

### Official Review · Reviewer_zzwD · 2023-10-30

**Soundness:** 3 good
**Presentation:** 3 good
**Contribution:** 2 fair
**Rating:** 6
**Confidence:** 4

**Summary:**

The authors investigate whether it is necessary to learn spatial convolutional filter weights in CNNs at all, to achieve good performance on data-specific tasks. Their approach is to fix the spatial convolutional kernel at initialization, only allowing learning linear combinations of these random spatial kernels during training. Authors argue that this approach has a regularising effect, as it prevents overfitting of the spatial kernels to specific spatial patterns occurring in the training data, also showing that learnable spatial convolutions in combination with a high rate of linear combinations actually decreases validation accuracy. Authors show that the benefits of learning linear combinations of random spatial kernels are mostly present at small kernel sizes, as for larger kernel sizes performance of the standard learnable spatial kernel and random spatial kernel starts to diverge. Authors conclude by giving a number of possible research extensions of their work, e.g. modifying the standard random i.i.d. initialization to reflect spatial distributions found in trained spatial kernels could be a very interesting next step.

**Strengths:**

Authors are very thorough in positioning their work and contrasting it with previous methods. I appreciate the thorough literature review, as it contributes to a clear context for the current work. The main research question that the authors investigate; the possibility of training a CNN without modifying the original spatial convolution weights, is well addressed theoretically and empirically, at least in the setting of image classification. The authors perform a number of relevant ablations over the hyperparameters of their method (i.e. kernel size, expansion rate) and provide extensive analysis of their results. Furthermore, where the method underperforms (larger kernel sizes), the authors give a qualitative analysis through visualisation of the distribution of learned kernels and provide pointers for possible ways of addressing this problem.

**Weaknesses:**

- My main objection with respect to this manuscript is the clarity of practical impact of this approach. It seems like in most settings, the proposed approach slightly underperforms traditional CNN architectures. For larger expansion rates, the model may outperform the baseline, but to me the trade-off in computational complexity is unclear. Although the author’s findings are interesting in their own right - it is certainly somewhat suprising that a factorization this drastic is still able to perform this impressively - I think the manuscript would greatly benefit from more clear indication of where potential benefits of frozen spatial convolution weights may lie, as the current version of the manuscript doesn’t sufficiently address this in my opinion.
- Second, the theoretical analysis of their approach seems somewhat limited. Although the authors list as one of their contributions the theoretical explanation for the impressive performance of their factorization, I'm not sure where in the paper this happens. On the one hand the authors say a large enough basis of random spatial filters provides enough expressivity to represent any other fully learnable spatial kernel (lemma 4.1), on the other hand they indicate that their approach has a clear regularizing effect, improving robustness. Are these not two contrasting observations?
- Third, the experiments conducted by the authors in the main body of the paper are quite limited in scope, in that they are all classification problems. Given that the authors themselves indicate that linear combinations of random weights have a hard time forming sharp spatial patterns, I think more attention should also be given to e.g. fine-grained segmentation (I’ve seen the results in the appendix but the authors might consider moving them to the main body and providing a more detailed experimental setup).

**Questions:**

- For the experiments in 5.1, how do the modified models and their baseline counterparts compare in terms of computational requirements? I.e. as you increase the expansion factor it increases computational complexity right? How do the different expansion factors and the original model compare, for example in terms of flops?
- In 5.3 the role of kernel size, you indicate that you investigate the spatial distribution for kernels for different kernel sizes. I assume the visualisations of the kernels shown for the frozen model are obtained after training? You indicate that there is a very clear spatial pattern in trained non-frozen kernels that isn’t present in the frozen model, and you indicate this as the main reason for trailing performance. This seems like a reasonable hypothesis (although somewhat in contrast to Fig 2), but would this mean your model is unable to perform more fine-grained tasks such as segmentation? Why not?
- Could you clarify what you see as the main practical impact of this approach? (see above)
- Could you clarify what you list as theoretical analysis for the performance of your approach? (see above)

---

> ### Author Response · Authors · 2023-11-13
> **Rebuttal 1/2**
>
> > W1:  [...]  practical impact of this approach [...]
> > Q3: Could you clarify what you see as the main practical impact of this approach? (see above)
>
> Our practical relevance is showing, that not learning the spatial convolutions obtains similar or even better accuracy/robustness in large networks. This comes at the benefit of a significantly reduced parameter count, and thus lower VRAM, faster training etc.
>
> *Are such large networks realistic?* Generally, we are all aware of the trend for larger models, but even today adversarial robustness SOTA (see https://robustbench.github.io/ under CIFAR-10,  $\ell_\infty$) uses WideResNets-70x16 [1] (16 times wider than a normal ResNet)! We assume that we would obtain cheaper and better training with freezing. Sadly we cannot verify this, due to the necessary GPU VRAM to run experiments.
>
> Just to clarify: generally, we do not promote training a frozen LCified network over a non-LC baseline. That would be indeed useless (although in some cases it is an easy trick to improve performance - e.g., see Tab. 4 in the Appendix). The LC-ResNets are just proxies for linear combinations in real architectures.
>
> Additionally, we deliver “food for thought”: i) pointwise convolutions after learnable spatial convolutions can impair performance - does this limit current architectures (e.g., ResNet-50)? ii) currently practiced initializations are not ideal when scaling kernel size (as commonly done) - spatial considerations are one thing to consider; iii) any studies of convolution filters must consider the presence of linear combinations.
>
> Does this clarify our intent? Would the reviewer suggest that we update our conclusion with this text?
>
> >W2.1: Second, the theoretical analysis of their approach seems somewhat limited. Although the authors list as one of their contributions the theoretical explanation for the impressive performance of their factorization, I'm not sure where in the paper this happens.
> Q4: Could you clarify what you list as theoretical analysis for the performance of your approach? (see above)
>
> We refer to the theory at the beginning of Sec. 4, where we show that a pointwise convolution following a spatial convolution is equal to a single convolution with a linear combination of the filters. Then it follows, that in such a scenario, freezing spatial filters to random inits and learning LCs is as expressive as learning them (if you have sufficiently many). Since random LC networks can learn approximations of fully learnable filters they perform similarly.
> We believe that this theory sufficiently explains why some networks can train to near-same accuracy with random spatial filters. Does this clarify the theoretical analysis?
>
> > W2.2: On the one hand the authors say a large enough basis of random spatial filters provides enough expressivity to represent any other fully learnable spatial kernel (lemma 4.1), on the other hand they indicate that their approach has a clear regularizing effect, improving robustness. Are these not two contrasting observations?
>
> Actually, they are not! The lemma only holds if you have sufficiently many kernels. In practice, you will have a limited number and only approximate the learnable filters (e.g., see Fig. 8 in the Appendix) - keep in mind, that the 1x1 operates on the $k\times k\times c_{in}$ filter and not kernel level and must create all $c_{out}$ filters ($k\times k\times c_{in}$) from the random filters.
>
> > W3: Third, the experiments conducted by the authors in the main body of the paper are quite limited in scope, in that they are all classification problems. Given that the authors themselves indicate that linear combinations of random weights have a hard time forming sharp spatial patterns, I think more attention should also be given to e.g. fine-grained segmentation (I’ve seen the results in the appendix but the authors might consider moving them to the main body and providing a more detailed experimental setup).
> Q2.3: … but would this mean your model is unable to perform more fine-grained tasks such as segmentation? Why not?
>
> Due to space limits, we decided to put these results in the appendix (as the trend is similar). Including these results would mean that we have to remove or shorten some other section. Do you have any suggestions on what we should replace?
>
> Regarding the sharp spatial patterns: This only affects larger kernels. Common models such as U-Net with 3x3 are not affected. Besides, “sharp” transitions in masks are generated by the high-frequency information on skip-connections, not the up-convolutions [1]. In fact, the results in the appendix are obtained on a U-Net.
>
> [1] Olaf Ronneberger, Philipp Fischer, Thomas Brox, “U-Net: Convolutional Networks for Biomedical Image Segmentation”, MICCAI, 2015.

---

> ### Author Response · Authors · 2023-11-13
> **Rebuttal 2/2**
>
> > Q1: how do the modified models and their baseline counterparts compare in terms of computational requirements?
>
> We deliberately did not report computational requirements, because we do not propose to use the LC-ResNets instead of off-the-shelf architectures (see above).
>
> Our message is freezing spatial filters reduces training time and does not affect accuracy (or even decreases) it in large networks. Here is an example of common networks. We report the number of trainable parameters (as a proxy for VRAM requirements) and latency (required time for one forward and backward pass). Please note that the inference time will depend on the framework, driver, hardware etc. FLOPs remain unaffected.
>
> | Model     	| Frozen 	| Trainable Parameters 	| Latency |
> |-----------|--------|----------------------|---------|
> | RN-50     	| no     	| 25.6 M               	| 141 ms  	|
> |           	| yes    	| 14.2 M               	| 131 ms  	|
> | WRN-50x2  	| no     	| 68.9 M               	| 211 ms  	|
> |           	| yes    	| 23.6 M               	| 185 ms  	|
> | WRN-101x2 	| no     	| 162.9 M              	| 359 ms  	|
> |           	| yes    	| 41.5 M               	| 311 ms  	|
>
> > Q2.1:  I assume the visualisations of the kernels shown for the frozen model are obtained after training?
>
> Yes, all visualized weights are shown after training.
>
> > Q2.2: You indicate that there is a very clear spatial pattern in trained non-frozen kernels that isn’t present in the frozen model, and you indicate this as the main reason for trailing performance. This seems like a reasonable hypothesis (although somewhat in contrast to Fig 2), …
>
> Fig. 2 shows a smooth resulting kernel - for an example of a “sharp” filter please see Fig. 8 in the Appendix. However, if you had enough filters/LCs, you could reproduce the sharp filters as such there is no contrast to Fig. 2.
>
> Please let us know if we have addressed all of your concerns. If you have any other questions/concerns please feel free to ask. We would be grateful if you would reconsider your score in light of our updates. Thank you again for your time and consideration.

---

### Official Review · Reviewer_6E1w · 2023-10-31

**Soundness:** 4 excellent
**Presentation:** 4 excellent
**Contribution:** 1 poor
**Rating:** 3
**Confidence:** 4

**Summary:**

The authors propose to investigate the impact of learned spatial filters v.s. random spatial filters in a deep Convolutional Neural Network (CNN) that uses 1x1 filters, the hypothesis being that the linear combinations (of the spatial filters) learned by 1x1 filters, are powerful enough themselves with random spatial filters alone to learn visual representations and attain good generalization on tasks such as image classification. The authors empirically investigate this hypothesis using existing CNN architectures with appropriate 1x1 filters, and furthermore by adding 1x1 filters to ensure that spatial filters are followed by a linear combination, to create "LC" blocks. The authors demonstrate that by drastically increasing the width of these LC blocks (i.e. both the spatial filters and 1x1), using only uniform randomly sampled spatial filters with learned 1x1 linear combinations, CNNs can match the generalization performance of using learned spatial filters on ResNet-based models with ImageNet, CIFAR-100, CIRAR-10, SVHN and Fashion-MNIST. The authors do also show however, a degradation in robustness performance when not using learned spatial filters that warrants further study.

**Strengths:**

* The paper is well-written overall, with a good background on both the recent developments in convolutional kernel sizes, and usage of 1x1 convolutions.
* The experimental setup is overall quite good overall (with some exceptions listed below), with the appropriate models and datasets used to demonstrate convincing empirical evidence on the author's hypothesis.
* The robustness results in the paper are perhaps the most novel/insightful part of the paper, and would benefit from more analysis/further study.

**Weaknesses:**

* The hypothesis and results are not surprising at all given how many linear combinations the authors learn in replacement of learned spatial filters. We already know that we can learn to reconstruct anything in a space by learning a linear combination of orthogonal basis vectors. While the basis vectors in this case are not designed to be orthogonal, they are sampled randomly sampled from a high-dimensional space (e.g. the kernel space of e.g. 3x3xC, where C is often very large). Randomly sampled vectors in high dimensional spaces are extremely likely to be orthogonal. Furthermore, we don't need to reconstruct perfectly any possible vector in filter space – the filters learned by CNNs likely represent a low-dimensional manifold within filter space, and approximating these filters rather than exact reconstruction is sufficient.
* Given the fact the hypothesis/results should not be surprising, this paper cannot just be considered an analysis paper pointing out a novel/surprising result to the research community, which is in itself a perfectly valid thing to do. The paper should be motivated outside of the fact that they hypothesis is true - i.e. is it useful to train a CNN in this manner over learning spatial filters? It's obvious that, as explained by the authors in the paper, using frozen LC models that match learned LC generalization performance are much more expensive in terms of compute and perhaps more importantly working memory (i.e. GPU VRAM), so this appears to not be the case at all.
* Many of the empirical results compare models with drastically different numbers of trainable parameters, while using the same training hyperparameters. However, we know that models with different numbers of trainable parameters usually need different hyperparameters and training setups. For example, models with more trainable parameters typically need longer to converge and require different learning rates, etc. Given this, it's not completely convincing that the model's generalization being compared between frozen lc and learned lc is fair.
* No explicit comparison of the compute and VRAM requirements for training models that are compared, in many cases there is quite a drastic difference it would seem. This is especially stark in section 5.1 where models with exponential increasing compute/param are compared with a fixed baseline model, i.e. ResNet-20-16.
* While I like the robustness analysis and think it warrants further attention, this is yet another reason why even when frozen LC models match generalization, they are poorly motivated.

**Questions:**

* If you believe this is not explained simply as learning from an orthogonal basis, as explained above, how many of the randomly sampled spatial filters are not orthogonal? You could perform an analysis to understand the representational ability of the random basis used.
* If you do believe this explanation, what does your paper contribute that is not simply empirical validation of this understanding? i.e. things that are specific to the DNN context, and other findings that are not trivially a result of being able to learn from an orthogonal basis.
* Why is it useful to train a frozen LC model v.s. a learned spatial model that achieve the same generalization? What is the tradeoff in compute and working memory (i.e. GPU VRAM) utilization for achieving this with frozen LC models v.s. a baseline learned spatial model for the same generalization performance?
* Did you do any hyperparameter turning for frozen LC v.s. learnable LC models you compared to ensure using the same hyperparameters/training setup is appropriate?

---

> ### Author Response · Authors · 2023-11-13
> **Rebuttal 1/2**
>
> > W1: The hypothesis and results are not surprising at all given how many linear combinations the authors learn in replacement of learned spatial filters. [...]
> > Q1: If you believe this is not explained simply as learning from an orthogonal basis, as explained above, how many of the randomly sampled spatial filters are not orthogonal? [...]
>
> The central assumption of the reviewer’s rejection rating regarding the orthogonality does not hold - neither in theory nor in practice! Theoretically (by the law of large numbers), the reviewer's orthogonality assumption only holds for very high dimensional vectors ( $C \rightarrow \infty$ ), but the actual dimensionality of C used in our evaluation (and realistic applications) is far from that. We have $ C \leq 64$ in our experiments.
> To verify this, we have followed your suggestion and computed this on the largest layer in a ResNet-LC20x32 under both, uniform and normal inits. This model breaks even with the baseline - as such it should contain a significant ratio of orthogonality. Yet, even when we conservatively define orthogonality with cosine similarity < 0.2, **0% of the filter pairs are orthogonal** in both cases.
>
> > W2: [...] is it useful to train a CNN in this manner over learning spatial filters? It's obvious that, as explained by the authors in the paper, using frozen LC models that match learned LC generalization performance are much more expensive [....]
> > Q2: what does your paper contribute that is not simply empirical validation of this understanding?
> > Q3.1: Why is it useful to train a frozen LC model v.s. a learned spatial model that achieve the same generalization?
>
> Just to clarify: Generally, we do not promote training a frozen LCified network over a non-LC baseline. That would be indeed useless (although in some cases it is an easy trick to improve performance - e.g., see Tab. 4 in the Appendix). The ResNets-LC are just proxies for linear combinations in real architectures.
>
> Instead, we ask: why would you train all parameters, if not learning the spatial convolutions obtains similar or even better accuracy at a significantly reduced parameter count and lower VRAM, faster training etc.? Our results show that, eventually, the utility of learned convolutions saturates and beyond that deteriorates performance - if networks are sufficiently large.
>
>
> *Are such networks realistic?* Yes. SOTA research on adversarial robustness (see https://robustbench.github.io/ under CIFAR-10,  $\ell_\infty$) already uses WideResNets-70x16 [2] (16 times wider than a normal ResNet)! We assume that we would obtain cheaper and better training with freezing. Sadly we cannot verify this, due to the necessary GPU VRAM to run experiments. And, generally, we are all aware of the trend for larger models.
>
>
> Does this clarify our intent? Would the reviewer suggest that we update our conclusion with this text?
>
> > W3: Many of the empirical results compare models with drastically different numbers of trainable parameters, while using the same training hyperparameters. [...]
> > Q4: Did you do any hyperparameter turning for frozen LC v.s. learnable LC models you compared to ensure using the same hyperparameters/training setup is appropriate?
>
> We have opted for an extra-long schedule relative to the small datasets (200 Epochs) to ensure appropriate convergence. We have ensured that the training accuracy of learnable LC-ResNets at any expansion is near 100%, i.e. they have converged. Further, earlier experiments with shorter schedules (75 Epochs under cosine decay) showed similar trends. Thus we are confident that we appropriately train the models. Tuning is mostly trial-and-error, of course, it may be possible to skew the results in either way - we are confident that using these parameters for all experiments allows for a fairer comparison.

---

> ### Author Response · Authors · 2023-11-13
> **Rebuttal 2/2**
>
> > W4: No explicit comparison of the compute and VRAM requirements for training models that are compared, in many cases there is quite a drastic difference it would seem. This is especially stark in section 5.1 where models with exponential increasing compute/param are compared with a fixed baseline model, i.e. ResNet-20-16.
> > Q3.2: What is the tradeoff in compute and working memory (i.e. GPU VRAM) utilization for achieving this with frozen LC models v.s. a baseline learned spatial model for the same generalization performance?
>
> We deliberately did not report computational requirements, because we do not propose to use the ResNet-LCs instead of off-the-shelf architectures (see above).
>
> Our message is freezing spatial filters reduces training time and does not affect (or even decreases) accuracy in large networks. Here is an example of common networks. We report the number of trainable parameters (as a proxy for VRAM requirements) and latency (required time for one forward and backward pass). Please note that the inference time will depend on the framework, driver, hardware etc.
>
> | Model     	| Frozen 	| Trainable Parameters 	| Latency |
> |-----------|--------|----------------------|---------|
> | RN-50     	| no     	| 25.6 M               	| 141 ms  	|
> |           	| yes    	| 14.2 M               	| 131 ms  	|
> | WRN-50x2  	| no     	| 68.9 M               	| 211 ms  	|
> |           	| yes    	| 23.6 M               	| 185 ms  	|
> | WRN-101x2 	| no     	| 162.9 M              	| 359 ms  	|
> |           	| yes    	| 41.5 M               	| 311 ms  	|
>
> > W5; While I like the robustness analysis and think it warrants further attention, this is yet another reason why even when frozen LC models match generalization, they are poorly motivated.
>
> We aim to explore the robustness aspect in future work. We have also observed that by using label smoothing we can further amplify the effect. One idea that we can give in this rebuttal is that due to the linear combinations, filters tend to be smoother, and at least for the initial layer this has been linked to improved robustness [1].
>
> [1] Haohan Wang, Xindi Wu, Zeyi Huang, Eric P. Xing, “High-frequency Component Helps Explain the Generalization of Convolutional Neural Networks”, CVPR, 2020.
>
> [2] ShengYun Peng, Weilin Xu, Cory Cornelius, Matthew Hull, Kevin Li, Rahul Duggal, Mansi Phute, Jason Martin, Duen Horng Chau, “Robust Principles: Architectural Design Principles for Adversarially Robust CNNs”, BMVC, 2023.
>
> **Please let us know if we have addressed your concerns. If you have any other questions/concerns please feel free to ask. We would be grateful if you would reconsider your score in light of our updates. Thank you again for your time and consideration.**

---

> > ### Author Response · Authors · 2023-11-20
> > **Follow-up with Reviewer 6E1w**
> >
> > Dear Reviewer, we would appreciate your feedback on our rebuttal. Could you kindly let us know if we addressed your concerns?

---

> ### Comment · Reviewer_6E1w · 2023-11-22
>
> I'd like to thank the authors for their rebuttal, and apologize for my late participation in the rebuttal period - this was due to exceptional circumstances.
>
> > The central assumption of the reviewer’s rejection rating regarding the orthogonality does not hold - neither in theory nor in practice! Theoretically (by the law of large numbers), the reviewer's orthogonality assumption only holds for very high dimensional vectors (  ), but the actual dimensionality of C used in our evaluation (and realistic applications) is far from that. We have  in our experiments.
> To verify this, we have followed your suggestion and computed this on the largest layer in a ResNet-LC20x32 under both, uniform and normal inits. This model breaks even with the baseline - as such it should contain a significant ratio of orthogonality. Yet, even when we conservatively define orthogonality with cosine similarity < 0.2, 0% of the filter pairs are orthogonal in both cases.
>
> This has little to do with the law of large numbers. It's just simply a fact that sampling random vectors from high-dimensional spaces is unlikely to result in vectors that are collinear, and yes even 3x3x64 is still a relatively high-dimensional space where I believe that will hold. If all your basis vectors (i.e. spatial filters) are co-linear as you are claiming above, you could never represent anything very different with any of your LC models's filters... so that makes very little sense! I believe it sounds like you have not computed the orthogonality of the spatial filters correctly (note this is the random spatial filters only you should be looking at).
>
> > Just to clarify: Generally, we do not promote training a frozen LCified network over a non-LC baseline. That would be indeed useless (although in some cases it is an easy trick to improve performance - e.g., see Tab. 4 in the Appendix). The ResNets-LC are just proxies for linear combinations in real architectures.
> > Instead, we ask: why would you train all parameters, if not learning the spatial convolutions obtains similar or even better accuracy at a significantly reduced parameter count and lower VRAM, faster training etc.?
> > ...
> >Are such networks realistic? Yes. SOTA research on adversarial robustness (see https://robustbench.github.io/ under CIFAR-10, ) already uses WideResNets-70x16 [2] (16 times wider than a normal ResNet)! We assume that we would obtain cheaper and better training with freezing. Sadly we cannot verify this, due to the necessary GPU VRAM to run experiments. And, generally, we are all aware of the trend for larger models.
> >
> > Does this clarify our intent? Would the reviewer suggest that we update our conclusion with this text?
>
> Unfortunately that doesn't clarify anything for me, in many ways it confuses me further. If your question is "why would you train all parameters..", and simultaneously you are claiming "we do not promote training a frozen LCified network over a non-LC baseline. That would be indeed useless", then hasn't that answered the question you claim to pose? Also since VRAM is the most bottlenecked resource for DNN training on GPU, any tradeoffs you show for "efficiency" in training LC models at the sacrifice of VRAM is very poorly motivated indeed.
>
> > Our results show that, eventually, the utility of learned convolutions saturates and beyond that deteriorates performance - if networks are sufficiently large.
>
> This is indeed the most interesting bit of your results I'll grant you if it holds up, but I'm not convinced that the results do hold up given the other issues I highlighted on using the same hyper-parameters across models of very different numbers of learnable parameters.
>
> > We have opted for an extra-long schedule relative to the small datasets (200 Epochs) to ensure appropriate convergence. We have ensured that the training accuracy of learnable LC-ResNets at any expansion is near 100%, i.e. they have converged. Further, earlier experiments with shorter schedules (75 Epochs under cosine decay) showed similar trends. Thus we are confident that we appropriately train the models. Tuning is mostly trial-and-error, of course, it may be possible to skew the results in either way - we are confident that using these parameters for all experiments allows for a fairer comparison.
>
> An extra-long training schedule does not make up for potentially poor choices of hyper-parameters in training models with very different numbers of trainable parameters. While I appreciate this is a difficult thing to do, giving us some idea of how you've found the hyper parameters for different models (or trying some tuning to verify they are reasonable) is necessary when you are comparing two training runs of very different models.
>
> Unfortunately this rebuttal has only made me question some of the analysis and results further. I apologize for being so late to the review process, but I hope the authors can further clarify.

---

> > ### Author Response · Authors · 2023-11-23
> >
> > Thank you for your response. Unfortunately, we are seeing this shortly before the deadline but we hope that we can clarify the remaining concerns anyway.
> >
> > > This has little to do with the law of large numbers. It's just simply a fact that sampling random vectors from high-dimensional spaces is unlikely to result in vectors that are collinear, and yes even 3x3x64 is still a relatively high-dimensional space where I believe that will hold. If all your basis vectors (i.e. spatial filters) are co-linear as you are claiming above, you could never represent anything very different with any of your LC models's filters... so that makes very little sense! I believe it sounds like you have not computed the orthogonality of the spatial filters correctly (note this is the random spatial filters only you should be looking at).
> >
> > Just because a basis isn't orthogonal, it doesn't imply co-linearity, either. A basis can be a basis, even if it is non-orthogonal.
> > We double-checked the code multiple times. Our computation of orthogonality is 100% correct. We have hard evidence against this claim.
> >
> > > [...] If your question is "why would you train all parameters..", and simultaneously you are claiming "we do not promote training a frozen LCified network over a non-LC baseline. That would be indeed useless", then hasn't that answered the question you claim to pose? [...]
> >
> > Sorry for the confusion. What we meant is that we do not suggest that anyone should replace spatial convolutions by LCBlocks as we did for our main study. Instead, we promote that when training large (wide) modern networks having these LCs by default, you can freeze spatial convolutions with little to no impact.
> >
> > For example, it makes little sense to take a Wide-ResNet-70x16 and switch the convolutions to LCBlocks. Instead, just freeze the spatial parameters - LCs are already present in this architecture (assuming it is a BottleNeck-ResNet).
> >
> > > This is indeed the most interesting bit of your results I'll grant you if it holds up, but I'm not convinced that the results ["the utility of learned convolutions saturates and beyond that deteriorates performance"] do hold up given the other issues I highlighted on using the same hyper-parameters across models of very different numbers of learnable parameters.
> > > An extra-long training schedule does not make up for potentially poor choices of hyper-parameters in training models with very different numbers of trainable parameters. While I appreciate this is a difficult thing to do, giving us some idea of how you've found the hyper parameters for different models (or trying some tuning to verify they are reasonable) is necessary when you are comparing two training runs of very different models.
> >
> > The hyperparameters for the *baseline* are well-tuned because we chose well-known parameters. For smaller datasets, these are the common CIFAR-10 parameters, for ImageNet, these are the parameters determined in timm [1]. Thus, in any case, the parameters for the *frozen random LC* experiments are okay. For the *learnable LC* we can only imagine non-convergence but the runs were all reaching 99+% training accuracy. So, there is no reason to be concerned beyond the usual difficulties of selecting the "correct" hyperparameters.
> >
> >
> > [1] Ross Wightman, Hugo Touvron, Hervé Jégou, "ResNet strikes back: An improved training procedure in timm", 2021.

---

### Official Review · Reviewer_SmiK · 2023-11-01

**Soundness:** 3 good
**Presentation:** 4 excellent
**Contribution:** 3 good
**Rating:** 6
**Confidence:** 3

**Summary:**

This paper analyzes the power of using random convolutions in different neural architectures. The authors find that linear combinations suffice to effectively recombine random conv filters into expressive network operations (by means of 1x1 convs). The authors also emphasize on the fact that the difference between learnable and random kernels becomes larger as the size of the kernels is increased.

**Strengths:**

The presentation of the paper is very good. All the findings are provided in a digestible and organized way, making the paper an interesting, intriguing and enjoyable read. The study is carried out in a systematic way, shedding light into the modus operandi of CNNs.

Altogether I believe this is a very strong submission with a little flaw in its evaluation.

**Weaknesses:**

The only weakness I see is that the experiments in Section 5.2 are not conclusive. I would encourage the authors to try and improve this part as it weakens the analysis of the paper –which is very strong up to this point.

**Questions:**

Do you have any insights wrt modus operandi of long convolutional models and how it differs from that of conventional (small kernel) architectures?

---

> ### Author Response · Authors · 2023-11-13
> **Rebuttal**
>
> > S1: Altogether I believe this is a very strong submission with a little flaw in its evaluation.
>
> Thank you for stating that our submission is very strong! Could you please clarify why you think the evaluation has a flaw? We would be grateful if you could tell us what we need to improve for you to increase your score.
>
> > W1: The only weakness I see is that the experiments in Section 5.2 are not conclusive. I would encourage the authors to try and improve this part as it weakens the analysis of the paper –which is very strong up to this point.
>
> Sec. 5.2 (“Scaling to other problems”) shows that our results are not limited to CIFAR-10 but scale to other datasets and tasks. We think it is a very important section - otherwise, readers may be concerned that the findings don’t scale (e.g., see Reviewer gGqG). Could it be that you meant to some other section?
>
> > Q1: Do you have any insights wrt modus operandi of long convolutional models and how it differs from that of conventional (small kernel) architectures?
>
> Long convolutions are more popular in NLP tasks, where it seems reasonable to assume that there is a correlation between distance in the context and relevance, i.e., far tokens are less relevant than closer ones. As such, long convolution kernels often show a decay of magnitude with depth (e.g., see [1]), which is very similar to the large kernel findings on vision we have in Sec. 5.2. Thus we would expect similar implications: I) current init methods won’t allow to us to reconstruct these filters only via LCs from random filters; II) we need better init methods for large/long convolution weights.
>
> [1] Y. Li, T. Cai, Y. Zhang, D. Chen, and D. Dey, “What makes convolutional models great on long sequence modeling?”, arXiv preprint arXiv:2210.09298, 2022.

---

> > ### Comment · Reviewer_SmiK · 2023-11-22
> >
> > Thank you for your answers.
> >
> > Wrt W1, I was referring to the final of Sec. 5.2. Here you stated that on ImageNet, "we were not able to outperform the baseline, but we attribute this to the granularity of the tested expansion rates and the fluctuations in the measurements of a single run", which gives the feeling that experiments on ImageNet were inconclusive. Could you comment on that?
> >
> > Best,
> >
> > The reviewer

---

> > > ### Author Response · Authors · 2023-11-22
> > >
> > > Thank you so much for your response!
> > >
> > > In contrast to the other experiments, we only trained 1 model for ImageNet (more expensive training). Thus, we might have gotten an unlucky seed (see [1]) for the non-baseline experiments. As you can see in Tab 2., there is only a marginal difference (0.14%) between the baseline and random frozen LCx64. There is no reason to believe that ImageNet is an outlier. On average (i.e., 3+ seeds) there will be no difference on ImageNet and random frozen will be as good or better.
> > >
> > > Additionally, the exact "sweet spot “ for this architecture/dataset may be at some odd value, e.g., 48 or 61. We didn’t think determining the exact value would be interesting, but we can run these experiments for the camera-ready version if the reviewer suggests this.
> > >
> > > *We would be grateful if you would consider increasing your score in light of our clarifications.*
> > >
> > >
> > > [1] David Picard, "Torch.manual_seed(3407) is all you need: On the influence of random seeds in deep learning architectures for computer vision", Preprint, 2021. https://arxiv.org/abs/2109.08203

---

### Author Response · Authors · 2023-11-13
**General response**

Thank you to the reviewers for their thoughtful feedback. All reviewers agree that our presentation is clear and technically sound. Reviewer SmiK calls our paper a "very strong submission" and Reviewer zzwD appreciates our “thorough literature review”, “extensive analysis”, and “well-addressed problem theoretically and empirically”.

However, our intentions and findings may have not been communicated clearly enough. As such, we would like to clarify them here, before addressing individual reviewer comments separately:

1. We do not propose to train LC-Blocks for practical applications. They are just proxies for our in-depth investigation of linear combinations which play an increasing role in modern CNN architectures and help us to study their effect systematically.
2.  Our theoretical and empirical investigation shows, that:
 A) linear combinations of Random frozen filters can outperform fully trainable baselines in clean and robust accuracy
B) In modern architectures, learned spatial convolution filters only marginally improve the performance compared to frozen random models in small kernel regimes. The reason for this is an implicit computation of linear combinations between pointwise and spatial filters.
C) Adding learnable pointwise convolutions immediately after spatial convolutions can decrease performance and robustness.
D) Only, under increasing kernel size, learned convolution filters begin to increase in importance but only due to a different spatial distribution of weights that cannot be reflected by currently practiced i.i.d. initializations.
3. From these findings, the following, novel and practically relevant conclusions and new open questions are arising:
	I) for any design of CNN architectures or investigation on CNN properties, the effect of linear combinations by 1x1 layers need to be taken into account
	II) the positive effects towards robustness induced by the linear combinations needs further investigation

**We would like to improve our submission with the valuable feedback of the reviewers and ask for their engagement in the discussion period and will provide an updated manuscript at the end of the discussion.**

We thank them again for their time and consideration.

---

### Meta-Review · Area_Chair_nML4 · 2023-12-16

**Metareview:**

This paper examines CNN architectures in which spatial convolution layers are frozen during training and are interleaved with learnable 1x1 convolutional layers.  Sufficiently expanding the set of frozen spatial filters allows such networks to perform on par with fully learnable baselines.

After the author response and discussion, reviewer opinion is split, with primary concerns being the degree to which these claims are surprising, as well as the scope of experimental validation.

Reviewer 6E1w comments that "the hypothesis and results are not surprising at all given how many linear combinations the authors learn in replacement of learned spatial filters."  Subsequent discussion with authors does not resolve this concern.  As the results in the paper are based on showing that wider frozen convolutional layers (more filters) are needed to match a smaller learnable baseline, the AC agrees with Reviewer 6E1w.

Reviewers zzwD and SmiK both raise concerns over the limited scope of the experiments.  Reviewer SmiK notes "experiments in Section 5.2 are not conclusive", with specific concern over ImageNet results.  The author response of "we only trained 1 model for ImageNet" does not assuage this concern.  Even though Reviewer SmiK gives a marginal accept rating, the AC believes inconclusive results on larger-scale datasets are a major issue that should be resolved.

**Justification For Why Not Higher Score:**

Unresolved reviewer concerns over whether or not the primary claim is trivial (unsurprising) and lack of sufficient experimental validation beyond small-scale datasets.

**Justification For Why Not Lower Score:**

N/A

---

### Decision · Program_Chairs · 2024-01-16

Reject